# Carbon Fiber-Reinforced Thermoplastic Composite Coatings for Steel Pipelines

**DOI:** 10.3390/polym16233417

**Published:** 2024-12-05

**Authors:** Ahmed I. A. Abd El-Mageed, Mohamed M. Desouky, Mamdouh El-Sayed, Tarek Salem, Ahmed Bahgat Radwan, Mohammad K. Hassan, Affaf K. Al-Oufy, Hassan M. El-Dessouky

**Affiliations:** 1Advanced Composites Research Lab, Faculty of Science, Galala University, Galala City 43511, Egypt; mohamed.mousa@gu.edu.eg (M.M.D.); mamdouh.elsayed@gu.edu.eg (M.E.-S.); 2Colloids & Advanced Materials Group, Chemistry Department, Faculty of Science, Minia University, Minia 61519, Egypt; 3Dyeing, Printing and Auxiliaries Department, Textile Research Institute, National Research Centre, Cairo 12622, Egypt; 4Center for Advanced Materials, Qatar University, Doha 2713, Qatar; ahmedbahgat@qu.edu.qa (A.B.R.); mohamed.hassan@qu.edu.qa (M.K.H.); 5Mechanical Engineering Department, Faculty of Engineering, Galala University, Galala City 43511, Egypt; affaf.aloufy@gu.edu.eg; 6Textile Engineering Department, Faculty of Engineering, Alexandria University, Alexandria 21544, Egypt; 7Physics Department, Faculty of Science, Mansoura University, Mansoura 35516, Egypt

**Keywords:** oil and gas, carbon fiber, thermoplastic composite, coating, steel pipelines

## Abstract

Steel pipeline systems carry about three-quarters of the world’s oil and gas. Such pipelines need to be coated to prevent corrosion and erosion. An alternative to the current epoxy-based coating, a multi-layered composite coating is developed in this research. The composite coatings were made from carbon fiber-reinforced thermoplastic polymer (CFRTP) material. Uniaxial carbon fiber CF/PPS prepreg tape was utilized, where the PPS (polyphenylene sulfide) is employed as a thermoplastic (TP) matrix. Compression molding was used to manufacture three flat panels, each consisting of seven plies: UD (Unidirectional), Biaxial, and Off-axis. Samples of carbon steel were coated with multi-layered composites. The physical, mechanical, and corrosion-resistant properties of steel-composite coated samples were evaluated. A better and more promising lap-shear strength of about 58 MPa was demonstrated. When compared to the Biaxial and Off-axis samples, the UD assembly had the maximum flexural strength (420 MPa); however, the Biaxial coating has the highest corrosion resistance (445 kΩ·cm^2^) when compared to the Off-axis and UD coatings.

## 1. Introduction

Carbon Fiber-Reinforced Thermoplastic (CFRTP) composites are increasingly gaining attention in various industries, such as aerospace, automotive, marine, and construction, due to their unique advantages over traditional thermoset composites. CFRTP composites offer a range of benefits, such as short consolidation cycles, shape forming, faster and lower-cost manufacturing, longer shelf life, reusability, recyclability, chemical and moisture resistance, as well as lightweight structures [1,2,3,4]. Such advantages enable CFRTP composites to have potential use in the oil and gas industry, particularly in pipeline applications. However, despite these advantages, CFRTP composites face difficulties that limit their widespread adoption in the energy sector, as improving the performance/cost ratio of CFRTP composites is still challenging.

Over the last 30 years, the protective coating technology for oil and gas pipelines has continued to evolve, addressing both technical and economic challenges with innovative solutions. As the industry seeks to enhance pipeline safety, efficiency, and longevity, advancements in materials used for coating formulations play a crucial role in overcoming the limitations of traditional coatings. The harsh environments in which these pipelines operate, coupled with the complex chemical and physical interactions between the coatings and their surroundings, make this a challenging area for research and innovation [5].

Pipelines are the major efficient means for the transportation of oil and gas around the world. Most of the world’s oil and gas is transported via metallic pipelines. However, they are subjected to erosion and corrosion under harsh employment conditions that can cause catastrophic failure. Different grades of carbon steel are being used; among them, high-strength low-alloy (HSLA) steels are extensively utilized to fabricate pipeline systems for oil and gas [6]. Therefore, corrosion inhibitors and protective coatings such as cathodic, inorganic, and polymer-based materials should be applied [7,8]. The most common types for coating pipelines are (i) Three-Layer of polyethylene/polypropylene (PE/PP), (ii) Fusion Bonded Epoxy (FBE), (iii) Coal Tar Enamel (CTE), and (iv) Asphalt Enamel and Polyurethane (PUR). All have limited chemical and heat resistance; therefore, improvement and/or development of alternatives are highly recommended to meet the demands of the oil and gas sector [9].

Specific characteristics must be fulfilled by the polymeric coatings in order to meet industry demands, such as good thermal, chemical, and photo-stability, high corrosion resistance, good mechanical strength, and low gas permeability. Despite the versatility, chemical resistance to corrosive media, and intrinsic toughness of traditional epoxy resins for the protection of steel pipelines, especially when combined with other protection procedures [10,11], the permeability to chloride ions, water, and oxygen gas causes hydrolytic degradation, adhesion loss, and coat deterioration. Hence, epoxy resins cannot successfully operate under high temperatures, pressures, and severe corrosive environments [12]. Therefore, different types of polymers are utilized to enhance corrosion protection efficiencies, such as unsaturated polyester, polystyrene, polyurethane, polyethyleneimine, and polysiloxane [13,14]. Nevertheless, the enhancement of protection performance is usually insufficient to prevent the coating layer’s detachment and deterioration of the underlying metal.

Each type of coating has advantages and disadvantages. Table 1 displays a specific comparison between one of the most common materials (PUR) used for coating in oil and gas pipeline systems and the thermoplastic (PPS) coating developed in this work [15,16,17,18,19,20,21].

To this end, several types of multi-layered coatings have been recently proposed to solve this issue and to enhance the performance of the polymer layer in harsh environments. Among these, pre-coating with a zinc underlying layer or an epoxy zinc-rich layer is used as a primer underneath the polymer. Although significant progress and extensive efforts have been exerted in this area of research, the cost and efficiency of the multi-layered composites are still below the level of commercialization and large-scale application [22].

However, some types of fiber-reinforced polymer thermoset composites are already used in industry as a coating for repairing cracked, damaged sections of steel pipes. For example, DEKOTEC GmbH [23] used carbon fiber composite (DEXPAND^®^-CF70) as a sleeve for a damaged section of steel pipe. Additionally, there is a case study reported by Advanced FRP Systems [24] to repair an internally corroded pipe using carbon fiber composite bands (CF-500 BD), where they initially applied weld seams on the damaged area of the steel pipe, then a multi-layer of CF bands was overlapped to cover the entire length of the repair. Critica Infrastructure (CSNRI) [25] has developed a similar technique to repair a cracked pipe using welded steel sleeves and a carbon fiber composite wrap. Therefore, this work proposes an alternative and innovative approach to multi-layered composite coating (carbon fiber-reinforced thermoplastic).

In this research, one of the main objectives is to find out how the improved structure/layup could be used as an external coating for the offshore and onshore pipeline systems, taking into consideration the environmental factors such as the corrosive ions in seawater and weather moisture, as well as extreme temperature conditions. Therefore, multi-layered coatings for protecting steel with superior mechanical and corrosion-resistant properties are developed. The CFRTP composite materials are used as a multi-layered coating. Unidirectional (UD) CF/TP prepreg tape is utilized. The polyphenylene sulfide (PPS) is a thermoplastic (TP) matrix. Three flat laminates, each made of seven plies: UD, Biaxial, and Off-axis, were manufactured using compression molding. The multi-layered composite coating is bonded to the outer surface of a carbon-steel substrate. The physical, mechanical, and chemical/corrosion tests were carried out. The results were compared to assess the enhanced mechanical and corrosion performance of the improved CFRTP composite coating.

## 2. Materials and Methods

### 2.1. Materials

Thermoplastic carbon fiber prepreg tape has been sourced from Hexcel Composites, Cambridge, UK (Figure 1). It is a 300 mm wide UD carbon fiber-reinforced thermoplastics (polyphenylene sulfide, PPS) material (CF/PPS). Table 2 lists the information about this CF/PPS TP prepreg as received.

In addition, a set of low carbon-steel plates (250 mm × 250 mm × 1 mm), grade DD11 (Appendix A), the same material used for manufacturing oil and gas pipelines, was provided by Ezz Steel, Cairo, Egypt.

Araldite^®^2011 structural adhesive (manufactured by HUNTSMAN, Bad Säckingen, Germany) is a multipurpose, two-component, room-temperature curing adhesive of high strength and toughness. It is suitable for bonding a wide variety of metals, ceramics, glass, rubber, rigid plastics, and most other materials in common use. It is a versatile adhesive for the craftsman and most industrial applications. A sample of its two components, Araldite^®^ 2011/A & Araldite^®^ 2011/B, has been provided by BODO MÖLLER CHEMIE, Cairo, Egypt (Appendix A), and its information as received is listed in Appendix A [26]. The resin/hardener mix (100 of A: 80 of B) may be applied manually or robotically to the pre-treated and dry joint surfaces. A layer of 0.05 to 0.10 mm thick adhesive will normally impart the greatest lap-shear strength to the joint. Huntsman stresses that proper adhesive joint design is critical for a durable bond. The joint components should be assembled and secured in a fixed position as soon as the adhesive has been applied.

### 2.2. Methods

#### 2.2.1. Composites Manufacturing

In terms of the surface roughness of the carbon steel, a mechanical method (angle hand grinder) was applied to increase the roughness of the surface prior to the bonding process. Three composite panels with different layups were manufactured using compression molding. Figure 2 (top-right corners) shows schematics for layering the three composites suggested.

Each panel (250 mm × 250 mm × 1 mm) consists of seven plies: Figure 2a shows the unidirectional (UD) panel, where all seven plies are oriented in one direction as [0,0,0,0,0,0,0]. Figure 2b shows the Biaxial one, where the seven plies are oriented in two directions, 0° and 90°, as [0,90,0,90,0,90,0]. Figure 2c shows the Off-axis one, where the seven plies are oriented in different directions as [0,45,−45,90,45,−45,0].

The CF/PPS composite samples were manufactured via compression molding using the following consolidation profile (Appendix A). The hot press conditions were programmed to apply temperature = 300 °C, pressure = 1 bar, and consolidation time = 10 min. Figure 2 below shows the consolidated CF/PPS composite panels obtained: (a) UD, (b) Biaxial, and (c) Off-axis samples.

#### 2.2.2. Metal-Composite Coating Bonding

The carbon-steel plates (250 mm × 250 mm × 1 mm) and CF/PPS composite panels (250 mm × 250 mm × 1 mm) were cut into small specimens (coupons) to obtain two sets of configurations: set (I) = 125 mm × 20 mm × 1 mm for mechanical tests and set (II) = 20 mm × 20mm × 1mm for burn-off and corrosion tests.

Figure 3 shows a selection of the coupon cut of steel and composite samples. According to the guide and data sheet of Araldite^®^ 2011 adhesive agent, 100 parts of Araldite^®^ 2011: Part A and 80 parts of Araldite^®^ 2011: Part B were mixed and ready to use immediately between steel and composite coatings. After applying the adhesive between the steel and composite coating, the coupons were assembled overnight to cure. Figure 4 shows a selection of metal-composite bonded coupons from the three samples: UD, Biaxial, and Off-axis, including images for the steel-composite interface for the Biaxial sample.

#### 2.2.3. Physical Characteristics: Burn-Off Test

Small coupons (set II) of composite panels were used to carry out the burn-off or burnout test [27] using a Muffle furnace with ceramic crucibles (see Appendix A). Five samples from each composite panel were heated to 600 °C for 30 min in the Muffle furnace. The polymer matrix (PPS) is burned, and the carbon fibers (CF) remain residual in the crucible. The crucible is weighed after 30 min of cooling down in a desiccator, and finally, the weight of the reinforcement CF is determined. Then, the fiber and matrix volume fractions and the density of the composites manufactured are calculated.

#### 2.2.4. Optical Microscopy

Specimens 20 mm × 20 mm (set II) from each bonded sample: UD, Biaxial, and Off-axis were used to prepare microsections. Microsections were mounted in resin via a Struers LaboPress-3 hot mounting machine. The specimens were prepared by initially grinding on SiC abrasive paper lubricated with water, using the following sequence of papers: P220, P320, P500, P1200, P2000, and P4000. The specimens were thereafter chemically and mechanically polished with a mixture of hydrogen peroxide (30% concentration) and colloidal silica solution, which contains 0.04 μm grit. The specimens were further etched with Kroll’s reagent, which is a mixture of 2 mL hydrofluoric acid (HF): 6 mL nitric acid (HNO_3_): 100 mL distilled water, for 20 s [28]. The microstructure of the polished specimens (Appendix A) was then optically scanned using an IM 500 (ECHO LAB) inverted microscope.

#### 2.2.5. Corrosion Test

The corrosion resistance of the synthesized thermoplastic composite coatings was evaluated in 3.5 wt.% NaCl using a Gamry 3000 potentiostat/galvanostat/ZRA (Warminster, PA, USA). A saturated Ag/AgCl electrode served as the reference electrode, while graphite was used as the counter electrode, and the composite coating functioned as the working electrode (see Appendix A). Electrochemical impedance spectroscopy (EIS) measurements were taken at an open-circuit potential with an AC signal of 10 mV amplitude over a frequency range of 10^5^ to 10^−2^ Hz. Throughout all corrosion assessments, the exposed area of the composite coating was kept at 0.785 cm^2^. All electrochemical tests were conducted at room temperature.

#### 2.2.6. Mechanical Test: Flexural (3-Point Bending)

A test method for flexural properties of FRP composite materials, Procedure A, ASTM D7264/D7264M-07 [29], is used to determine the flexural strength of steel-composite samples manufactured in this research. The flexural test is beneficial for identifying the layup errors in the case of multi-layered laminates and for determining the tensile force applied to the bottom (steel) of the test sample and the compressive force applied to its top (CFRTP coating). This means the CFRTP coating is always in compression during flexural strength testing. Appendix A shows the universal testing machine used for conducting the flexural tests for the steel-composite bonded samples: UD, Biaxial, and Off-axis. Five coupons (repeats) of each sample are tested.

#### 2.2.7. Mechanical Test: Single-Lap Shear

Lap shear is mainly designed to determine the Lap-Shear Strength (LSS) of adhesives for bonding two similar or dissimilar materials together using an adhesive agent. In this study, the ASTM D5868-01 test [30] is used to determine the adhesive strength between steel and composite coatings. Appendix A shows the setup for the single-lap-shear test with a steel-composite bonded sample. The adhesive (bonded) area is 20 mm × 30 mm, and the length of each part (composite and steel) is 110 mm.

## 3. Results and Discussion

In order to examine the quality and integrity of composite coatings manufactured in this study, the fiber volume fraction (V_f_) was determined using the matrix burn-off test. Five specimens were tested from each composite configuration. The densities of carbon fiber (CF), Polyphenylene sulfide (PPS), and CF/PPS composite are 1.8, 1.35, and 1.6 g/cm^3^, respectively. Table 3 gives the results obtained from the burn-off test for the three manufactured composites: UD, Biaxial, and Off-axis.

It is expected that the density of the manufactured composites varies due to the compression molding process, as some fibers may be washed out when the resin or matrix is subjected to excessive shear stress. By comparing the results obtained in Table 3 with those received in Table 2, the Biaxial composite panel exhibited improved properties, followed by the Off-axis one. However, around 50% of carbon fibers are washed out from the Off-axis laminate.

The optical cross sections tend to confirm the layup and configuration of the three composite laminates as manufactured. Figure 5 shows a selection of the optical micrographs obtained for the composite microsections. The round cross sections of CF indicate the 0° direction (Figure 5a), the elliptical ones indicate the ±45° direction (Figure 5c), and the longitudinal sections correspond to the fibers in the 90° direction (Figure 5b), as highlighted in the micrographs. In Figure 5a, the black spots refer to the voids/porosity formed due to air trapped during the manufacturing process of composite samples. The void content looks significant in the case of the UD sample, which is already confirmed by the results of the burn-off test (Table 3).

As an example, Figure 6 highlights the interfacial boundaries between the carbon-steel adhesive agent (A) and composite coating-adhesive layer (B) in the Biaxial sample. Interface A depicts the surface roughness applied to the steel before bonding to ensure strong adhesion between the two dissimilar materials. On the other hand, a smooth adhesion between the two similar materials (adhesive and composite) is highlighted by interface B. Interfaces A and B confirm that the adhesion between steel and composite coating is strong and delamination-free (Figure 7).

The main aim of this research is to develop the CFRTP composite coating for carbon steel to improve corrosion resistance in the case of offshore and onshore conditions. Figure 8 presents Nyquist plots for the carbon steel coated with CFRTP composites: (a) UD, (b) Biaxial, and (c) Off-axis coatings. To characterize the electrochemical behavior of these coatings, the EIS data were modeled using the electrical equivalent circuits, as depicted in Figure 9.

The equivalent circuit shown in Figure 9a fits the EIS data for both Biaxial and Off-axis coatings. In contrast, the EIS data for the UD coating were modeled using the equivalent circuit depicted in Figure 9b. The electric circuits consist of R_s_, representing the resistance of the brine solution used in the test. At the same time, R_c_ and R_ct_ correspond to the composite coating resistance and the charge transfer resistance of the coatings, respectively. Constant phase elements (CPE1 and CPE2) were employed instead of pure capacitors to account for the non-ideal behavior at the surface and interface of the metallic coatings. This approach is justified by the following equation [31,32].
(1)$$Z_{CPE}=\frac{1}{Q{\left(j\omega \right)}^{n}}$$

$Z_{CPE}$ is the impedance of the constant phase element, j is the imaginary unit, *ω* is the angular frequency, and *n* is the phase shift, which indicates the deviation from ideal capacitive behavior (with *n* ranging from 0 to 1). This model represents the real electrochemical system by accommodating surface roughness and inhomogeneity at the coating/electrolyte interface.

The double-layer capacitance was calculated using the following formula [33,34]:(2)$$C_{dl}=\sqrt[{n}]{\frac{Q}{{R_{x}}^{\left(\alpha -1\right)}}}$$

Q is the CPE constant, α is the CPE exponent, respectively. R represents the resistance.

It is evident that the Biaxial coating exhibits the highest corrosion resistance, measured at 445 kΩ·cm^2^, compared to 226 kΩ·cm^2^ for the Off-axis coating and 48 kΩ cm^2^ for the UD coating. This performance can be attributed to the symmetric architecture of carbon fibers (0°, 90°) inside the laminate as well as the reduction of voids and defects in the Biaxial coating. In the case of UD and Off-axis layup, the corrosive ions can easily find more channels to flow through the fiber interspacing into the steel substrate. In contrast, in the biaxial architecture (0°, 90°), the fibers are oriented in two directions, reducing the possible flow channels and thereby significantly delaying the penetration of corrosive ions.

The CPE exponents n_1_ and n_2_ reflect surface and interface homogeneity, respectively, which are closely related to corrosion behavior. Higher n_1_ values (e.g., 0.788 for Biaxial) indicate smoother, more uniform surfaces, reducing localized corrosion initiation, while lower n_1_ (e.g., 0.656 for UD) suggests increased roughness or defects, promoting corrosion. Similarly, higher n_2_ values (e.g., 0.847 for UD) signify a more homogeneous charge transfer interface, enhancing corrosion resistance. In contrast, lower n_2_ (e.g., 0.752 for UD) implies more significant interface irregularities, facilitating electrochemical activity and corrosion processes.

On the other hand, higher values of CPE1 and CPE2, such as 32 and 26 µ s^n^ Ohm^−1^ cm^−2^ for UD, suggest increased pseudo-capacitance, which can result from more active sites on the surface. These active sites are often associated with structural features like pores, defects, or increased roughness, facilitating charge accumulation and electrochemical reactions. In contrast, lower values (e.g., 4.6 and 0.02 µ s^n^ Ohm^−1^ for Biaxial) imply fewer defects or a smoother, less reactive surface, enhancing corrosion resistance.

Correspondingly, the Biaxial coating also showed the lowest double-layer capacitance (*C*_*d**l*_) of 7 nF, which correlates with its enhanced corrosion resistance. In contrast, the Off-axis and UD coatings exhibited significantly higher capacitance values of 63 nF and 36 nF, respectively. A lower *C*_*d**l*_ suggests a smaller effective surface area for electrochemical reactions or indicates a more robust coating that minimizes ion penetration. The reduced presence of imperfections in the Biaxial coating further hinders the ingress of aggressive species, such as chloride ions (Cl^−^), thus providing better protection for the metal surface against corrosion.

On the other hand, the Warburg phenomenon, observed in the EIS data of the UD coating (Figure 8a), occurs due to the diffusion-controlled process within the electrochemical system. This typically happens when the movement of ions or reactants through the electrolyte to the electrode surface is hindered, leading to frequency-dependent impedance. In the case of the coatings studied, the Warburg impedance indicates that the diffusion of corrosive species through the coating material plays a significant role in the overall corrosion resistance. This can be attributed to the composition of the coatings, which affects the porosity and permeability, thereby influencing the diffusion rates of the reacting species.

The Warburg impedance, which is typically represented as a straight line at 45 degrees in the Nyquist plot, reflects the diffusion of ionic species through the coating and into the substrate. In the Nyquist plot for the UD coating, the shape of the curve can be analyzed to understand the implications of Warburg diffusion. If the plot displays a semi-circle at higher frequencies, followed by a straight line at lower frequencies, this is characteristic of a diffusion-limited process. The straight line indicates that the impedance is primarily governed by the diffusion of corrosive species, such as chloride ions, through the porous structure of the coating. The Warburg impedance was measured at 18 µΩ cm^2^s^−1/2^, indicating a moderate resistance to the diffusion of hydrated chloride ions through the coating. When the carbon fiber coating is exposed to NaCl, the situation becomes more specific, as NaCl introduces hydrated chloride ions (Cl^−^) that play a critical role in the corrosion of carbon steel. In this case, the anodic half-reaction on the carbon steel would likely involve the oxidation of the metal (Fe → Fe^2+^ + 2e^−^), while the cathodic half-reaction could involve oxygen reduction (O_2_ + 2H_2_O + 4e^−^ 4OH^−^). The Warburg element in this scenario reflects the diffusion of hydrated chloride ions (Cl^−^) and metal cations (Fe^2+^) through the electrolyte. If the carbon fiber coating has defects or porosity, the hydrated chloride species can penetrate it, leading to localized corrosion, such as pitting. In this case, the Warburg impedance reflects the slow diffusion of these ions through the coating and the electrolyte, where mass transport becomes rate-limiting at lower frequencies. The rate-determining step would be the diffusion of chloride ions to the steel surface.

In contrast, the Biaxial and Off-axis coatings may not show this distinct Warburg behavior in their EIS data due to their structure diminishing diffusion pathways, resulting in different equivalent circuit models. The absence of a Warburg element in their circuits suggests that their corrosion resistance is primarily dictated by charge transfer processes rather than diffusion limitations. Thus, the shape of the Nyquist plot for the UD coating not only highlights the presence of Warburg diffusion but also reinforces the need for a tailored equivalent circuit to accurately model its unique electrochemical behavior.

In order to evaluate the mechanical performance of the steel-composite assembly, the 3-point bending (flexural) test was carried out. It is recommended that the failure mode of the tested samples should be identified to serve as a guide for recognizing localized damage that could impact the performance of real-world pipeline systems. There are six types of failure modes that can be observed during the 3-point bending test [35]: (F1) tensile fracture of fiber, (F2) tensile fracture at the outermost layer, (F3) compressive fracture, (F4) tensile fracture including interlaminar shear, (F5) compressive fracture including interlaminar shear, and (F6) delamination and interlaminar shear fracture. Appendix A shows a schematic demonstrating the possible six flexural failure modes.

Figure 10 shows a selection of the UD-tested samples and failure modes observed for each coupon. The two failure modes, F1 and F2, are the most predominant, and some of the samples failed due to adhesive debonding, which is almost similar to the failure mode F6 (delamination). It is expected to obtain the two modes F1 and F2 due to the layup of the sample, with all fibers laid in one direction, and the load applied across the fiber (out of the plane), i.e., there is no fiber in the other directions (90°) to resist such load.

The same behavior was obtained for the Biaxial samples; the failure modes F2, F3, and F6 were also observed, as shown in Figure 11. The Off-axis samples exhibited different failure modes, as shown in Figure 12. In addition to the failure mode F6 (adhesive debonding or delamination), there is a different type of failure called fiber wrinkling, followed by possible F1, which is difficult to see. In the Off-axis samples, the fibers are oriented in different directions (0°, ±45°, 90°); consequently, the applied load has been distributed in the same directions, resulting in the expected wrinkling failure mode. By comparing the three samples tested, the Off-axis coating seems to have the best load resistance and bonding strength.

Quantitatively, Figure 13 shows the average flexural strength for three steel-composite bonded samples. However, the UD sample presented the highest flexural strength (420 ± 28 MPa) compared to the Biaxial (186 ± 58 MPa) and Off-axis (365 ± 15 MPa) samples. The Off-axis sample exhibited the most consistent test repeats with the lowest standard deviation (±15°). The obtained value of the flexural strength for the UD sample (420 MPa) is relatively low compared to a previously reported value (1393 MPa) for CF/PPS without steel [21]. This is attributed to the higher void content (Table 3) as well as the presence of another dissimilar material (i.e., steel). A similar claim is also reported to confirm this point but for a different matrix [36]. The strength values obtained reflected the composite layup, as the Biaxial laminate should exhibit half of the strength of the UD sample, and the strength of the Off-axis should exhibit a value in between those of the Biaxial and the UD laminates.

A single-lap-shear test is conducted to determine the lap-shear strength (LSS) of the adhesive used for bonding steel and composite coating. Figure 14 demonstrates the tested coupons of (a) UD, (b) Biaxial, and (c) Off-axis samples. It was noticed that the adhesive agent is significantly elongated in the case of the UD sample compared to the Biaxial and Off-axis samples. This is highlighted by a red box in Figure 14. Such elongation occurs due to the presence of a possible channel along the fibers in the UD layup. In contrast, in the Biaxial and Off-axis architectures, the adhesive is constrained by the fibers in all directions.

The lap-shear test is considered a tensile test, and it is known that the UD composites should exhibit the highest tensile strength. This has been confirmed by the results obtained and shown in Figure 15, in which the UD sample presented an LSS of 58.4 ± 9.4 MPa, followed by the Off-axis of LSS = 28 ± 4 MPa and then the Biaxial of LSS = 20 ± 6 MPa.

According to the technical data sheet provided by HUNTSMAN [26], the adhesive LSS of steel-to-steel is around 25 MPa, and CFRP-to-CFRP (Carbon Fiber Reinforced Polymer Composite) is around 19 MPa. Here in this work, the LSS obtained for steel-composite coating is much better and more promising. In summary, the results obtained for corrosion resistance and mechanical tests of the developed composite coatings in this work are listed in Table 4. It is concluded that the UD samples exhibited improved flexural and lap-shear strengths compared to the Biaxial and Off-axis ones. However, regarding corrosion resistance, which is the main objective of this project, the Biaxial composite coating demonstrated the highest.

## 4. Conclusions

The following remarks can be concluded as follows:I.The appropriate technique for manufacturing thermoplastic composite panels is compression molding. Three CF/PPS composite plaques with UD, Biaxial, and Off-axis layups were produced.II.The performance of the developed composite coating was successfully assessed using the suggested multi-layered structure. There is good agreement between the results of physical tests through optical microscopy and the burn-off test. Both verified that, compared to the biaxial and off-axis samples, the UD laminate has a higher void content.III.The Biaxial coating has the strongest corrosion resistance, measuring 445 kΩ·cm^2^, while the Off-axis coating has 226 kΩ·cm^2^ and the UD coating has 48 kΩ cm^2^. This is despite the fact that the UD samples showed the highest flexural and lap-shear strengths when compared to the Biaxial and Off-axis ones.IV.In the following phase of this work, prototypes of steel pipes with different diameters will be manufactured as demonstrators for oil and gas applications. These pipes will be coated with the Biaxial CFRTP composite developed in this paper to protect them against high pressure and extremely high temperatures.

## Figures and Tables

**Figure 1 polymers-16-03417-f001:**
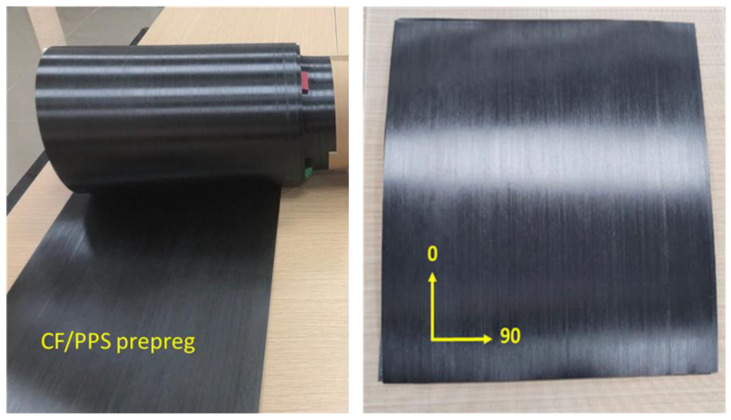
CF/PPS UD prepreg tape (**left**) and a single ply (**right**) were cut to show the fiber direction.

**Figure 2 polymers-16-03417-f002:**
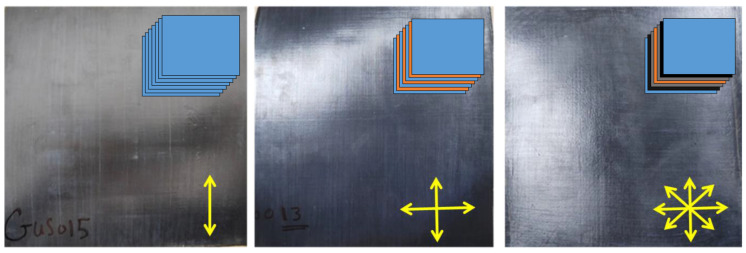
Conslidated CF/PPS of composite panels: (**a**) UD, (**b**) Biaxial, and (**c**) Off-axis.

**Figure 3 polymers-16-03417-f003:**
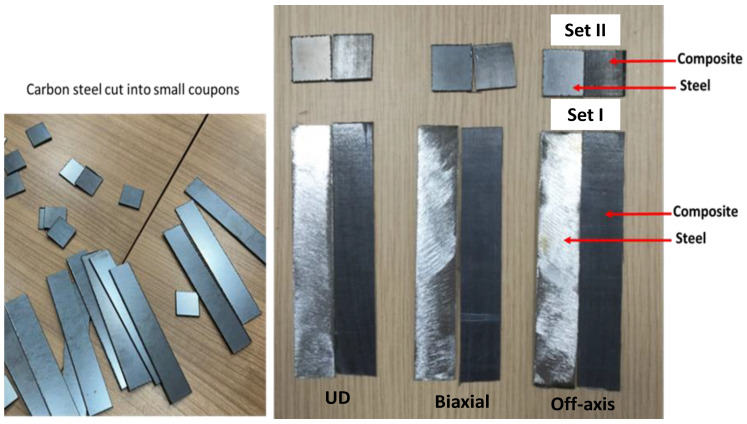
A selection of steel and composite coupons: set (I) and set (II).

**Figure 4 polymers-16-03417-f004:**
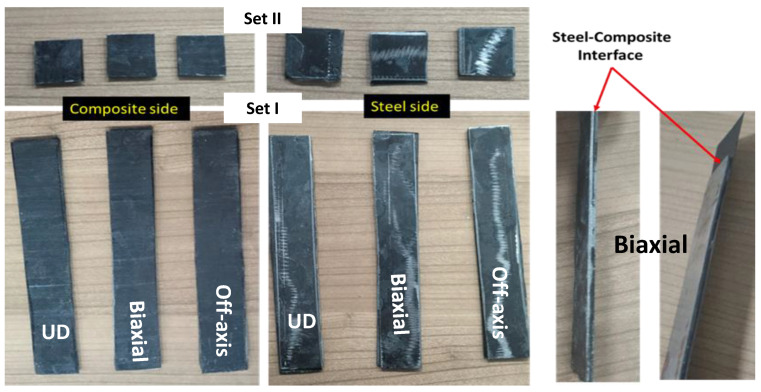
A selection of steel-composite bonded coupons from the three samples UD, Biaxaial, and Off-axis (**Left** and **Middle**). An example of steel-composite interface, Biaxial (**Right**).

**Figure 5 polymers-16-03417-f005:**
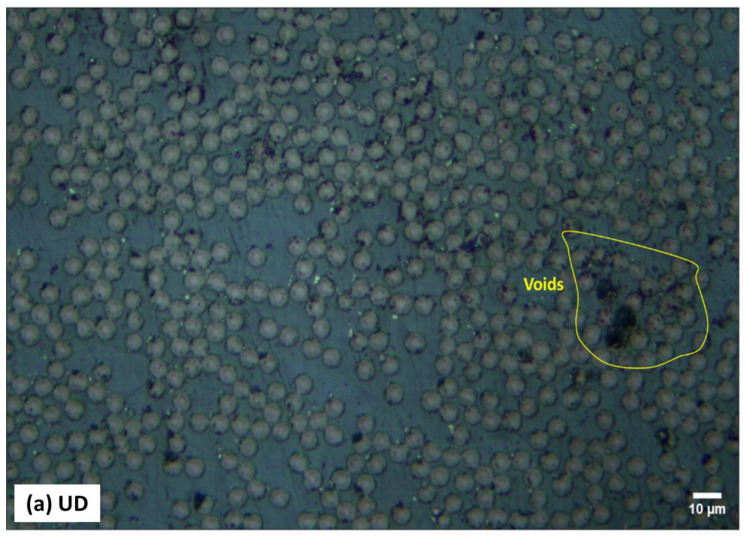
Microsections of three composite coatings: (**a**) UD, (**b**) Biaxial, and (**c**) Off-axis.

**Figure 6 polymers-16-03417-f006:**
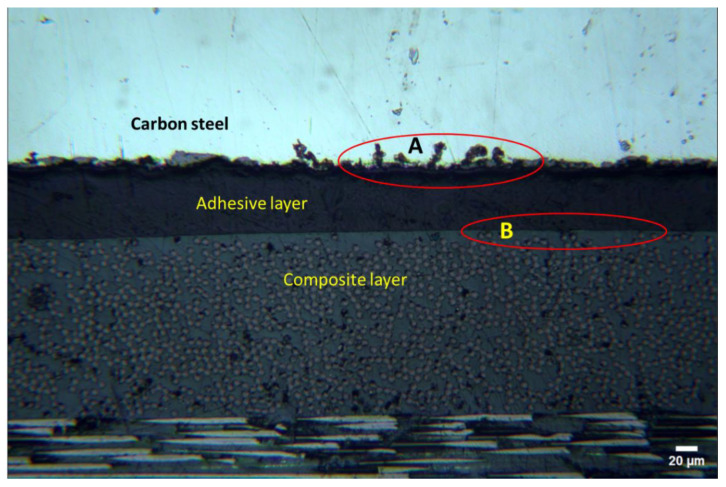
Interfacial boundaries between steel, adhesive, and composite materials in Biaxial sample.

**Figure 7 polymers-16-03417-f007:**
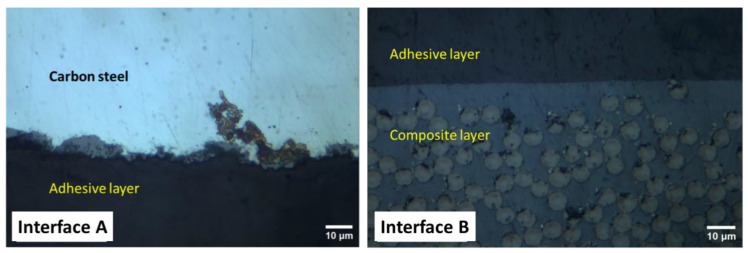
Zoomed-in sections of the interfaces A and B are highlighted in Figure 6.

**Figure 8 polymers-16-03417-f008:**
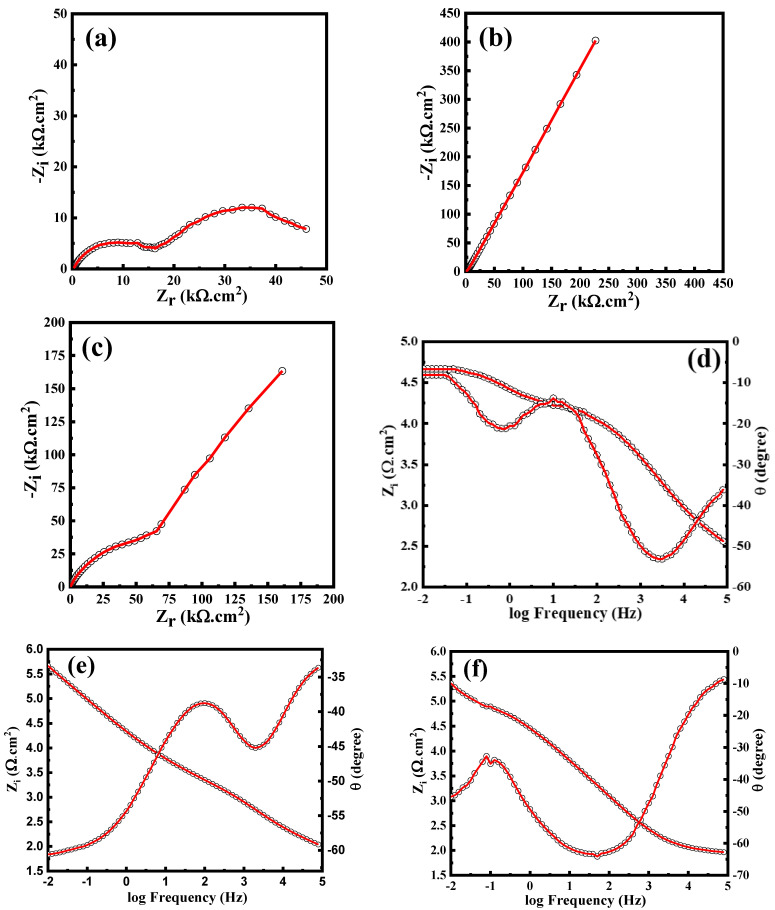
Nyquist plots of the (**a**) UD, (**b**) Biaxial, and (**c**) Off-axis in 3.5 wt.% NaCl and their relative Bode plots (**d**–**f**), respectively. The red solid line represents the fitting.

**Figure 9 polymers-16-03417-f009:**
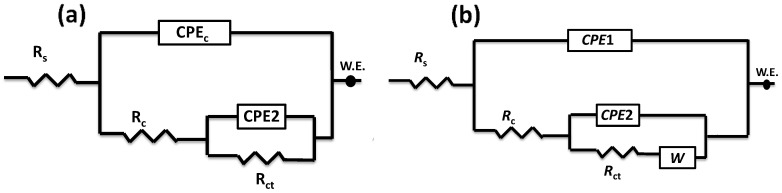
Equivalent electric circuits were applied to fit the measured EIS data for (**a**) Biaxial and Off-axis coatings, and (**b**) UD coating.

**Figure 10 polymers-16-03417-f010:**
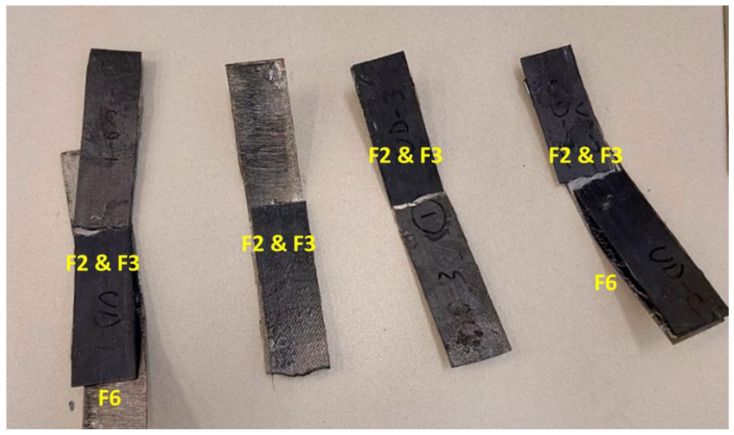
A selection of the UD-tested samples and failure modes observed.

**Figure 11 polymers-16-03417-f011:**
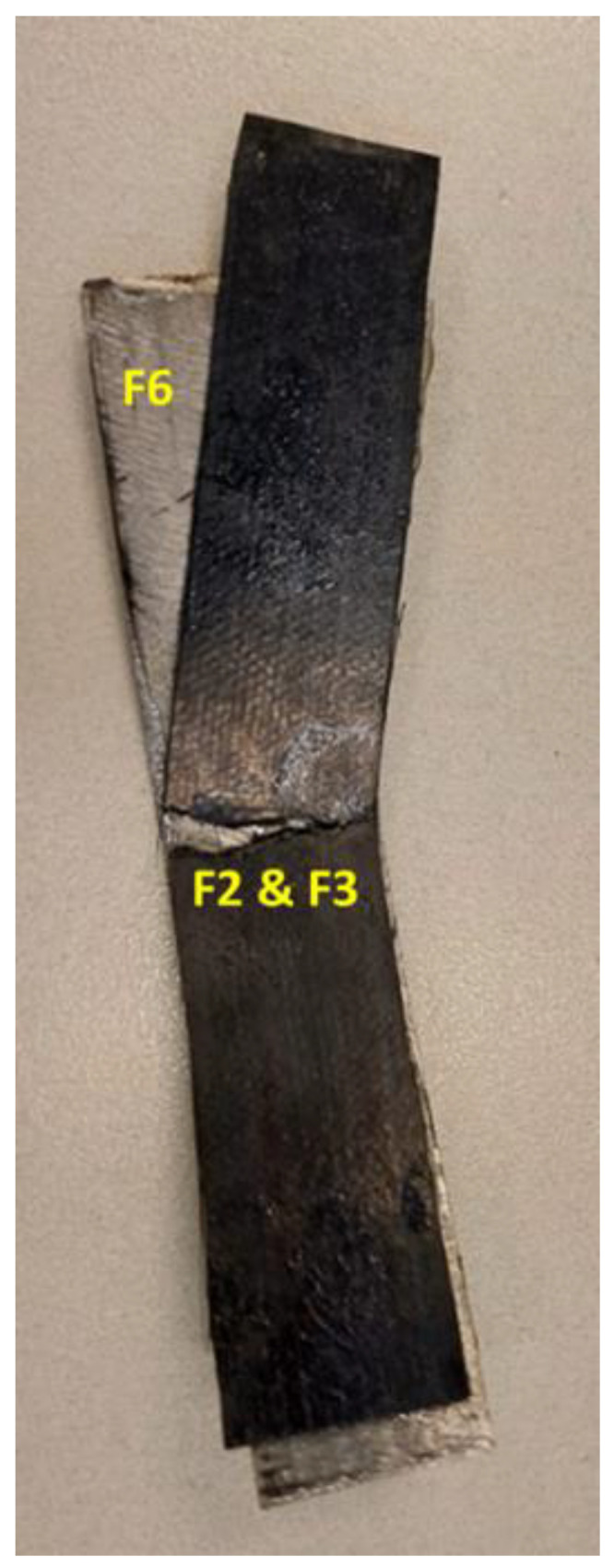
An example of the Biaxial tested sample and failure modes observed.

**Figure 12 polymers-16-03417-f012:**
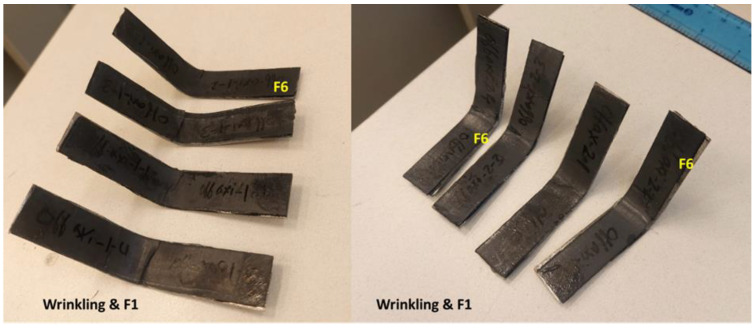
A selection of the Off-axis tested samples and failure modes was observed.

**Figure 13 polymers-16-03417-f013:**
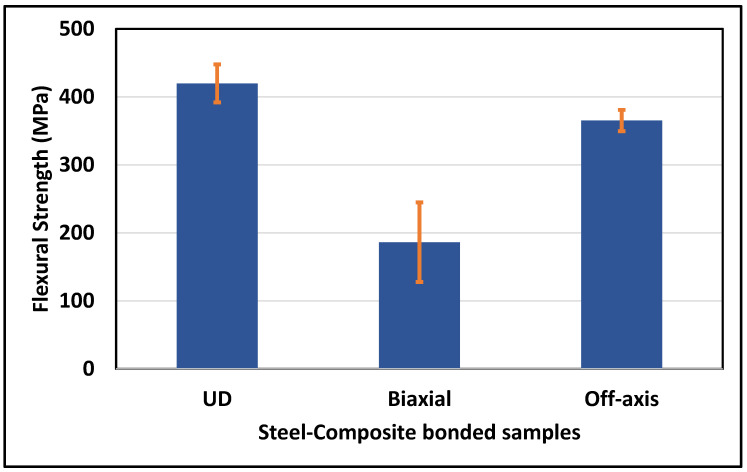
Flexural strength of steel-composite bonded samples.

**Figure 14 polymers-16-03417-f014:**
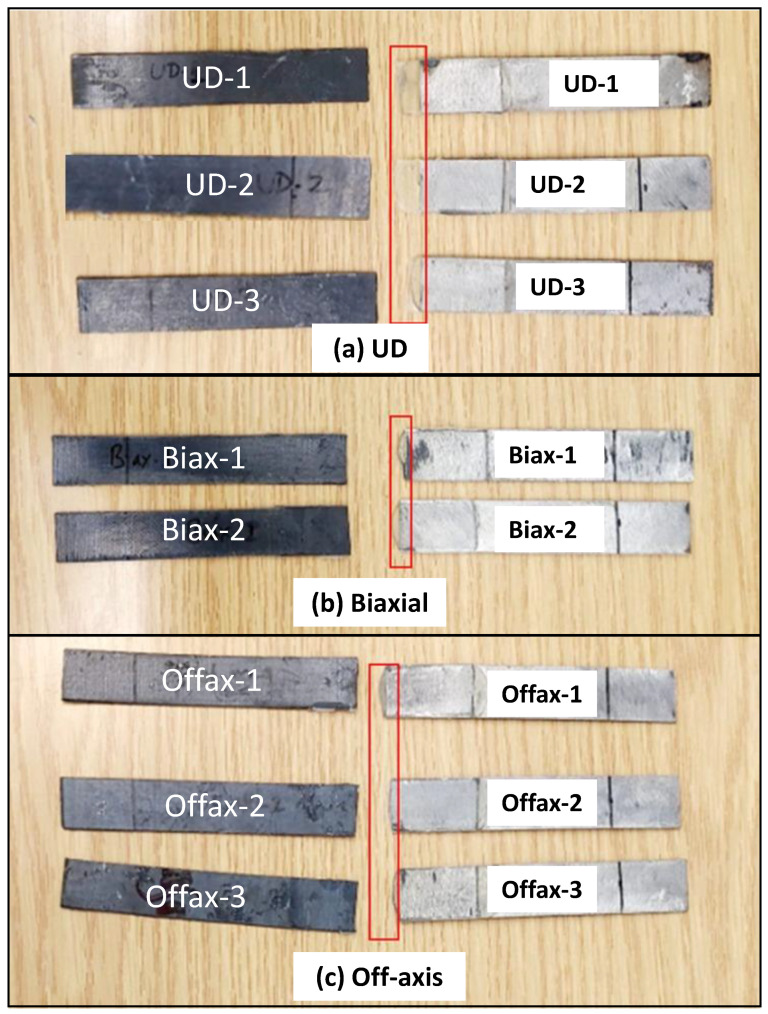
A selection of tested coupons that fell apart after the lap-shear test: (**a**) UD, (**b**) Biaxial, and (**c**) Off-axis.

**Figure 15 polymers-16-03417-f015:**
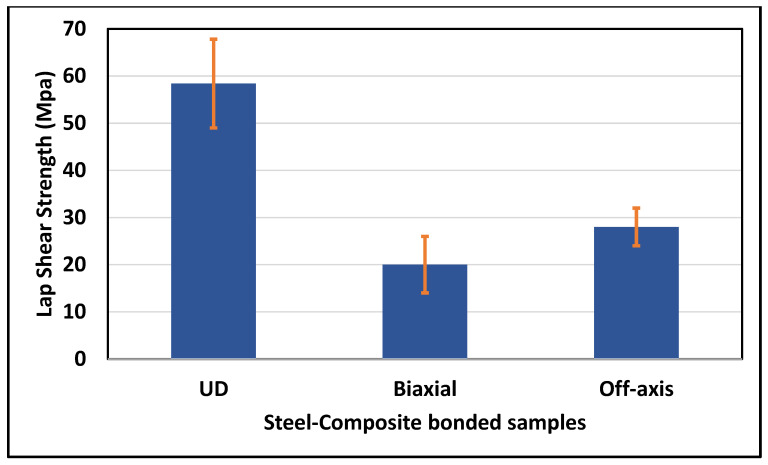
Lap-shear strength of steel-composite bonded samples.

**Table 1 polymers-16-03417-t001:** Comparison between one of the most common coatings used (PUR) and the thermoplastic (PPS) developed in this work.

Property	PUR	PPS
Cost	variable cost range based on grade and/or performance.	low-price and cost-effective
Recycling	recyclable	recyclable
Versatility	can be processed into various forms and shapes	easy forming and fast processing speed
Mechanical Performance	good mechanical properties	superior mechanical properties
Density	significant Lightweight (1.25 g/cm^3^)	lightweight (1.35 g/cm^3^)
Melting Temperature	Up to 188 °C	280 °C
Insulation Properties	excellent thermal and acoustic insulation	excellent chemical resistance
Flammability	highly flammable	non-flammable
Chemical Concerns and Health Risks	cause health risks	non-toxic material
Durability Over Time	corrosion resistance, as well as maintaining its shape and supportive properties over time	high temperature and corrosion resistance
Degradability	degradable and discolor over time when it exposures to sunlight and an extremely hot environment	Non-degradable, but the color can be changed over time

**Table 2 polymers-16-03417-t002:** Properties of CF/PPS UD prepreg as manufactured.

Product Name	No. Filaments in a Tow	CF Volume Fraction (%)	Resin Content (%)	Areal Weight (g/m^2^)	Ply Thickness (mm)	Density (g/cm^3^)
AS4C/PPS (TC1100)	12,000	60	40	227	0.15	1.6

**Table 3 polymers-16-03417-t003:** Results of burn-off test of composite samples ± STDEV.

Sample	V_f_ (%)	V_m_ (%)	Density (g/cm^3^)	Voids (%)
UD	45 ± 0.06	55 ± 0.06	1.552 ± 0.03	3.0
Biaxial	59 ± 0.06	41 ± 0.06	1.614 ± 0.03	0.0
Off-axis	33 ± 0.08	77 ± 0.04	1.609 ± 0.09	0.0

**Table 4 polymers-16-03417-t004:** Summary of the results obtained for the three composite coatings.

Sample/Property	Corrosion Resistance (kΩ·cm^2^)	Flexural Strength (MPa)	Lab Shear Strength (MPa)
UD	48	420	58.4
Biaxial	445	186	20
Off-axis	226	365	28

## Data Availability

All data generated or analysed during this study are included in this manuscript and its Appendix A.

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
