# Peer review of "Carbon Fiber-Reinforced Thermoplastic Composite Coatings for Steel Pipelines"

_polymers, 2024, doi:10.3390/polym16233417_

Round 1
Reviewer 1 Report (New Reviewer)
Comments and Suggestions for Authors
Here are my suggestions for enhancing the manuscript:
-
The article addresses a significant challenge in the oil and gas industry - corrosion and mechanical degradation of carbon steel pipelines. Highlighting the practical impact of these findings in the introduction could enhance the manuscript’s relevance.
-
The structure is clear, but sections like Materials and Methods would benefit from additional details, particularly about specific experimental techniques. Including brief explanations of technical terms and testing methods would make it more accessible for a broader audience.
-
Methodology is well-documented, especially for corrosion testing. Adding explanations for selected parameters (e.g., temperature, duration in burn-off tests) would improve reproducibility.
- Suggestion: Mention possible limitations or sources of variability, such as differences in composite layups or environmental factors.
-
Data presentation is clear, with organized figures, though restructuring some tables for readability (especially for comparing UD, Biaxial, and Off-axis layups) could help.
- Suggestion: Adding statistical analysis in the lap shear and flexural strength tests would strengthen reliability. A visual summary (e.g., bar chart of key performance metrics) could enhance clarity and comparisons.
-
The study’s findings on the superior corrosion resistance of the Biaxial coating are promising. Expanding the explanation for why this configuration outperformed others (e.g., structural factors influencing resistance) would provide deeper insight.
- Suggestion: Discuss potential applications of each composite configuration for varying pipeline environments.
-
Failure mode analysis during flexural testing is detailed and valuable. Expanding on how these failure modes could impact real-world pipeline performance would strengthen practical applicability.
- Examining the effects of void content on mechanical strength and durability would add to the discussion.
-
The conclusion effectively summarizes findings but could be enhanced by mentioning specific industry applications. For future research, consider exploring high-pressure environments or extreme temperature conditions.
-
References are current and relevant, capturing recent advancements. Including studies on alternative corrosion-resistant coatings for pipelines could broaden the context and deepen the background.
Overall, the manuscript presents a comprehensive and valuable study on CFRTP composite coatings for pipeline protection. Implementing these suggestions could enhance the manuscript’s clarity, relevance, and impact.
Comments on the Quality of English LanguageThe quality of English is generally good, but some improvements are recommended for clarity. Certain technical terms and complex sentences could be simplified or rephrased to make the content more accessible to a broader audience. Minor grammatical corrections would also enhance readability.
Author Response
Authors’ Response to Reviewers’ Comments
Manuscript ID. polymers-3305126
Authors would like to kindly thank the journal editorial team and reviewers for sharing such valuable comments. Of course, considering these comments will improve the manuscript and make it more powerful. Please find below the authors’ response to the suggested comments (point-by-point) highlighted in blue.
The changes will be added to the revised manuscript and highlighted in blue as well.
Reviewer #1
- The article addresses a significant challenge in the oil and gas industry - corrosion and mechanical degradation of carbon steel pipelines. Highlighting the practical impact of these findings in the introduction could enhance the manuscript’s relevance.
Thank you very much. The last paragraph in the introduction section (Page 5, Lines 103-114) summarises the practical key findings which highlight the impact of application of the multi-layered composite coating for pipelines systems made of steel such as the corrosion resistance and mechanical performance of CFRTP developed in this research.
- The structure is clear, but sections like Materials and Methods would benefit from additional details, particularly about specific experimental techniques. Including brief explanations of technical terms and testing methods would make it more accessible for a broader audience.
The authors appreciate the reviewer comment, we’d like to clarify that most of the details that the reviewer suggested to be added in the materials and methods section are mentioned in the supplementary file (Figures S3-S8). Unfortunately, we can’t add all the supplementary figures to the manuscript body due to the figures number limitations within the manuscript. The explanation of the technical terms and testing methods are also clearly presented in the text as well as the cited references (Standard methods i.e. 27, 29, 30, 35).
- Methodology is well-documented, especially for corrosion testing. Adding explanations for selected parameters (e.g., temperature, duration in burn-off tests) would improve reproducibility.
Thanks for the reviewer for appreciating the well-documentation of the Methodology section. In fact, the parameters (e.g., temperature, duration) of the burn off test are clearly mentioned in page 9 (Line 184).
- Suggestion: Mention possible limitations or sources of variability, such as differences in composite layups or environmental factors.
- Suggestion: Discuss potential applications of each composite configuration for varying pipeline environments.
Thanks for the suggestions. Please allow us to answer these two related points as follows:
In terms of the differences in the composite layups and their applications, authors have suggested the most common layup and architecture can be developed using a unidirectional prepreg tape (i.e. UD, Biaxial and Off-axis) as mentioned and demonstrated in page 7 (Figure 2). Regarding the environmental factors, the authors have added a new paragraph to the revised manuscript in page 5, Lines 103-107, “one of the main objectives is to find out the improved a structure/layup could be used as external coating for the off-shore and on-shore pipelines taking into consideration the environmental factors such as the corrosive ions within the sea water and weather moisture, as well as extreme temperature conditions, therefore”
- Data presentation is clear, with organized figures, though restructuring some tables for readability (especially for comparing UD, Biaxial, and Off-axis layups) could help.
It is a good point, so the authors have added a new sentence and a new Table (Table 4) to the revised manuscript in page 23, Lines 425 – 429“In summary, the results obtained for corrosion-resistance and mechanical tests of the developed composite coatings in this work are listed in Table 4. It is concluded that the UD samples exhibited the improved flexural and lap-shear strengths compared to the Biaxial and Off-axis ones. However, regarding the corrosion resistance, which is the main objective of this project, the Biaxial composite coating demonstrated the highest.”
Table 4. Summary of the results obtained for the three composite coatings.
|
Sample/property |
Corrosion rsistance (kΩ·cm²) |
Flexural strength (MPa) |
Lab shear strength (MPa) |
|
UD |
48 |
420 |
58.4 |
|
Biaxial |
445 |
186 |
20 |
|
Off-axis |
226 |
365 |
28 |
- Suggestion: Adding statistical analysis in the lap shear and flexural strength tests would strengthen reliability. A visual summary (e.g., bar chart of key performance metrics) could enhance clarity and comparisons.
Of course, adding statistical analysis to the manuscript is a good suggestion, but since we’ve just studied only three independent variables (UD, Biaxial and Off-axis layups), therefore, the statistical analytic model wouldn’t be reliable. However, the provided tables and bar charts in the manuscript are adequate to present the comparison between the properties of the different CFRTP composite coatings.
- The study’s findings on the superior corrosion resistance of the Biaxial coating are promising. Expanding the explanation for why this configuration outperformed others (e.g., structural factors influencing resistance) would provide deeper insight.
Thanks for the reviewer for appreciating the corrosion findings. In fact, the explanation for why Biaxial coating shows promising corrosion resistance are clearly mentioned in page 16.
- Failure mode analysis during flexural testing is detailed and valuable. Expanding on how these failure modes could impact real-world pipeline performance would strengthen practical applicability.
The authors appreciate the reviewer comment, we’d like to clarify the detailed figure of the different failure modes is provided in the supplementary file (Figure S9). In addition, the explanation of such modes is highlighted in the manuscript body in pages 19, 20 (Figures 10, 11 and 12).
Regarding how these failure modes could impact real-world pipeline performance, knowing such failure mode in the pipe will be used as a guide for identifying the localised damage. To consider this point, the authors have added a new sentence to the revised manuscript in page 18, Lines 367-369 “It is recommended that the failure mode of the tested samples should be identified to serve as a guide for recognizing localized damage that could impact the performance of real-world pipeline systems.”
- Examining the effects of void content on mechanical strength and durability would add to the discussion.
In fact, the effects of void content on mechanical strength and durability are clearly mentioned in page 20 (Lines 400-404) and in the conclusion section as well.
- The conclusion effectively summarizes findings but could be enhanced by mentioning specific industry applications. For future research, consider exploring high-pressure environments or extreme temperature conditions.
Thanks for the reviewer for appreciating conclusion. To take this point into consideration, the authors have added a new sentence to the revised manuscript in page 24, Lines 443-446 “In the following phase of this work, prototypes of steel pipes with different diameters will be manufactured as demonstrators for oil & gas applications. These pipes will be coated with the Biaxial CFRTP composite developed in this paper to protect them against high pressure and extremely high temperatures.”
- References are current and relevant, capturing recent advancements. Including studies on alternative corrosion-resistant coatings for pipelines could broaden the context and deepen the background.
The authors have already cited a reasonable number of references (references 5-14) discussing a wide range of corrosion-resistant coatings studies.
- The quality of English is generally good, but some improvements are recommended for clarity. Certain technical terms and complex sentences could be simplified or rephrased to make the content more accessible to a broader audience. Minor grammatical corrections would also enhance readability.
By taking the reviewers recommendation into consideration, the authors have checked the manuscript very carefully and corrected any possible typos and grammatical errors in the manuscript. Additionally, the authors have sent the manuscript to professional English language editing service to check the manuscript as well.
- Overall, the manuscript presents a comprehensive and valuable study on CFRTP composite coatings for pipeline protection. Implementing these suggestions could enhance the manuscript’s clarity, relevance, and impact.
Thank you very much for the valuable comments. We’ve implemented all the suggestions into the revised manuscript which becomes more powerful.

Reviewer 2 Report (New Reviewer)
Comments and Suggestions for Authors
Thank you for involving me for the reviewing of the manuscript entitled "Carbon Fibre-Reinforced Thermoplastic Composite Coatings for Steel Pipelines".There are some issues that need to be answered.
Major problems:
1) 5) The paper looks more like a report than a rigorous academic research paper. The author needs to make significant revisions to the paper.
2) The language of the entire paper needs to be improved significantly. A professional polishing is strongly recommended.
3) According to the description in Section 2.1, the author obtained three types of thermoplastic carbon fiber pre preg tape from external sources. Next, the author cut them into pieces and bonded them onto iron plates using purchased adhesive agent. Then, relevant experiments were conducted. The research depth and theoretical value of the entire paper are insufficient, and it is not recommended to publish it in Polymers.
4) As a research paper, many of the words in it are very informative and unnecessary, e.g. the second and third paragraphs in Section 2.2.5.
Other problems:
1) Keyword selection and sorting are unreasonable.
2) Figure 2: The design and layout of the figure is unreasonable.
3) Figure 14: Subheadings of subgraphs are reversed.
4) How can the author ensure the consistency of the sample surface after polishing with an angle grinder? This is very obvious in Figure 4. The impact of such consistency on the bonding results cannot be ignored.
5) Conclusions: The conclusion needs to be revised and streamlined.
6) The format of references is also inconsistent with the format of this journal. Suggest revising the paper according to the latest published papers.
Comments on the Quality of English LanguageA professional polishing is strongly recommended.
Author Response
Authors’ Response to Reviewers’ Comments
Manuscript ID. polymers-3305126
Authors would like to kindly thank the journal editorial team and reviewers for sharing such valuable comments. Of course, considering these comments will improve the manuscript and make it more powerful. Please find below the authors’ response to the suggested comments (point-by-point) highlighted in blue.
The changes will be added to the revised manuscript and highlighted in blue as well.
Reviewer #2
Thank you for involving me for the reviewing of the manuscript entitled "Carbon Fibre-Reinforced Thermoplastic Composite Coatings for Steel Pipelines". There are some issues that need to be answered.
Major problems:
- The paper looks more like a report than a rigorous academic research paper. The author needs to make significant revisions to the paper.
Sorry, taking into consideration reviewers’ suggestions and addressing all comments received, authors have revised the manuscript. The revised version has been significantly improved and hope it satisfies reviewers and editor.
- The language of the entire paper needs to be improved significantly. A professional polishing is strongly recommended.
Authors have checked the manuscript very carefully and corrected any typos and grammatical errors in the text. In addition, a professional English language editing service has polished the English of the text. The entire text of the revised manuscript has been improved.
- According to the description in Section 2.1, the author obtained three types of thermoplastic carbon fiber pre preg tape from external sources. Next, the author cut them into pieces and bonded them onto iron plates using purchased adhesive agent. Then, relevant experiments were conducted. The research depth and theoretical value of the entire paper are insufficient, and it is not recommended to publish it in Polymers.
Thanks for the comment, authors have only obtained the CF/PPS prepreg tape as a single sheet or ply from overseas as shown in Figure 1. Then 3 composite panels of different layups were fabricated using press moulding as detailed in section 2.2.1 Figure 2 and Figure S3 (in supplementary file). Then samples of each panel have been bonded to a steel substrate using adhesive agent Figure S2 (in supplementary file). Then 4 main experiments have been conducted and presented in depth in the manuscript as follows:
- Burn-off test & optical microscopy: sections 2.2.3 & 2.2.4
- Corrosion test: section 2.2.5
- Flexural test: section 2.2.6
- Lap-shear test: section 2.2.7
After implementing suggestions and addressing comments raised by reviewers, the revised version of the manuscript has been significantly improved and become more powerful ready for publication in Polymers, hopefully.
- As a research paper, many of the words in it are very informative and unnecessary, e.g. the second and third paragraphs in Section 2.2.5.
We agree with the reviewer's suggestion all unnecessary information are removed from the text.
Other problems:
- Keyword selection and sorting are unreasonable.
The set of keywords is amended in the revised version.
- Figure 2: The design and layout of the figure is unreasonable.
Sorry about that, the design of Figure 2 has been improved in the revised version.
- Figure 14: Subheadings of subgraphs are reversed.
Sorry about that, the subheadings of Figure 14 have been corrected in the revised version.
- How can the author ensure the consistency of the sample surface after polishing with an angle grinder? This is very obvious in Figure 4. The impact of such consistency on the bonding results cannot be ignored.
Sorry for the confusion, the steel side shown in Figure 4 is the outer surface, which is not bonded to composite. The other side of the steel bar, which is not shown in Figure 4 and shown in Figure 3 (set II, shiny surface), is the one gets bonded to composite using adhesive agent. We agreed with the reviewer that keeping the consistency of the surface roughness is very important; therefore, we did our best to do a uniform surface roughness using the angle grinder to ensure better adhesion between the two dissimilar surfaces.
- Conclusions: The conclusion needs to be revised and streamlined.
Agreed, the conclusion has been rewritten to be streamlined
- The format of references is also inconsistent with the format of this journal. Suggest revising the paper according to the latest published papers.
We have followed the author guidelines of the journal and the current references format (style) is the same as recommended by the journal.
- A professional polishing is strongly recommended.
Authors have checked the manuscript very carefully and corrected any typos and grammatical errors in the text. In addition, a professional English language editing service has polished the English of the text. The entire text of the revised manuscript has been improved.

Reviewer 3 Report (New Reviewer)
Comments and Suggestions for Authors
This study develops carbon fibre-reinforced thermoplastic polymer composite materials to manufacture a multi-layered coating for corrosion protection of pipelines. In my view, the research contains some novel and important results that may be of interest to the Journal's readers. However, I have several comments and questions that the authors should address before the manuscript can be recommended for publication.
1. The detailed description of the role, functions, and importance of the three-electrode system in EIS measurements, counter electrode, etc. (in subsection 2.2.5, Corrosion test) appears unnecessary for a specialist in the field of electrochemical corrosion, as these are well-known concepts. This section leaves a somewhat simplistic impression, and these paragraphs should be shortened, keeping only specific information and removing general descriptions at the level of student textbooks.
2. In addition to EIS measurements, classic electrochemical behavior tests using linear and/or cyclic voltammetry should also be conducted to evaluate the corrosion behavior and protective properties.
3. It is commendable that the authors used the concept of a constant phase element (CPE) to interpret the EIS measurements. However, the manuscript lacks results from numerical modeling. A table (possibly in the supplementary materials) should include the numerical values of both parameters, Q and n, for each constant phase element. A detailed discussion of these values is necessary, as they could provide valuable additional information on the kinetics and mechanism of the corrosion-electrochemical process.
4. The authors should explain how the double-layer capacitance (Cdl) was determined, as using a constant phase element in modeling electrochemical impedance excludes direct measurement of double-layer capacitance. In such cases, Cdl can be calculated using special theoretical models available in the literature.
5. The parameters of the Warburg element need quantification. It is acknowledged that this element appears under conditions of slow diffusion of specific particles (components) in the solution, but which particles exactly? At least an assumption should be made. In this context, a brief discussion of the chemistry of the corrosion process under study is necessary: what constitutes the anodic and cathodic half-reactions, what is the mechanism of the process, and which step is rate-determining?
6. The chemical composition of the solution used as a model corrosion environment is not specified. What dissolved species (component) did it contain (NaCl?) and at what concentration? Was dissolved oxygen removed, or was the solution under natural aeration conditions (the presence of dissolved oxygen affects the corrosion mechanism)?
7. The results of this study should be compared with prototypes and analogs described in the literature.
Author Response
Authors’ Response to Reviewers’ Comments
Manuscript ID. polymers-3305126
Authors would like to kindly thank the journal editorial team and reviewers for sharing such valuable comments. Of course, considering these comments will improve the manuscript and make it more powerful. Please find below the authors’ response to the suggested comments (point-by-point) highlighted in blue.
The changes will be added to the revised manuscript and highlighted in blue as well.
Reviewer #3
This study develops carbon fibre-reinforced thermoplastic polymer composite materials to manufacture a multi-layered coating for corrosion protection of pipelines. In my view, the research contains some novel and important results that may be of interest to the Journal's readers. However, I have several comments and questions that the authors should address before the manuscript can be recommended for publication.
- The detailed description of the role, functions, and importance of the three-electrode system in EIS measurements, counter electrode, etc. (in subsection 2.2.5, Corrosion test) appears unnecessary for a specialist in the field of electrochemical corrosion, as these are well-known concepts. This section leaves a somewhat simplistic impression, and these paragraphs should be shortened, keeping only specific information and removing general descriptions at the level of student textbooks.
We do agree with the reviewer. The paragraph has been shortened.
- In addition to EIS measurements, classic electrochemical behavior tests using linear and/or cyclic voltammetry should also be conducted to evaluate the corrosion behavior and protective properties.
In the case of our carbon fibre coating, we added an insulating adhesive layer to ensure proper adhesion between the substrate and the coating and to prevent galvanic corrosion. This adhesive layer, being non-conductive, prevents direct electronic transfer between the coating and the underlying material. As a result, CV is unsuitable for directly testing the corrosion behavior of non-conductive coatings due to the fundamental reliance of CV on electrical conductivity to generate a measurable current response as a function of applied voltage. Non-conductive coatings act as electrical insulators, impeding both ionic and electronic transfer between the underlying metallic substrate and the surrounding electrolyte. This effectively blocks the electrochemical reactions necessary for detecting corrosion-related processes, such as anodic dissolution or cathodic reduction, making it impossible to assess the coating's protective performance or degradation behavior using CV. Therefore, we employed EIS, as it allows for evaluating the coating's barrier properties by measuring the system's impedance over a range of frequencies without requiring a direct current flow through the non-conductive material.
- It is commendable that the authors used the concept of a constant phase element (CPE) to interpret the EIS measurements. However, the manuscript lacks results from numerical modeling. A table (possibly in the supplementary materials) should include the numerical values of both parameters, Q and n, for each constant phase element. A detailed discussion of these values is necessary, as they could provide valuable additional information on the kinetics and mechanism of the corrosion-electrochemical process.
The fitting parameters derived from the EIS measurements are listed in Table S2 in the supplementary file. A detailed discussion of these values is included in the manuscript and highlighted in blue (Pages 16-17).
- The authors should explain how the double-layer capacitance (Cdl) was determined, as using a constant phase element in modeling electrochemical impedance excludes direct measurement of double-layer capacitance. In such cases, Cdl can be calculated using special theoretical models available in the literature.
The calculated double layer has been calculated using the following formula [33,34]:
Q is the CPE constant, α is the CPE exponent, respectively. R represents the resistance.
Added to Page 16 in the revised manuscript includes two new references [33] and [34]
- The parameters of the Warburg element need quantification. It is acknowledged that this element appears under conditions of slow diffusion of specific particles (components) in the solution, but which particles exactly? At least an assumption should be made. In this context, a brief discussion of the chemistry of the corrosion process under study is necessary: what constitutes the anodic and cathodic half-reactions, what is the mechanism of the process, and which step is rate-determining?
Thank you for the clarification. We have added the value of Warburg element to Table S2 which is almost 18 µ Ω cm2s-1/2. This value indicates a moderate resistance to the diffusion of ions through the coating. Given that chloride ions are primarily responsible for the initiation of corrosion in chloride-rich environments, the rate-determining step in this case would be the diffusion of chloride ions through the coating to the underlying substrate. This is where the Warburg element becomes crucial, as it models the diffusion process and helps quantify the coating’s ability to protect the metal. When the carbon fibre coating is exposed to NaCl, the situation becomes more specific, as NaCl introduces hydrated chloride ions (Cl⁻) that play a critical role in the corrosion of carbon steel. In this case, the anodic half-reaction on the carbon steel would likely involve the oxidation of the metal (Fe → Fe²⁺ + 2e⁻), while the cathodic half-reaction could involve oxygen reduction (O₂ + 2H₂O + 4e⁻ → 4OH⁻) or the reduction of chloride ions in some cases. However, chloride ions are known to be aggressive in corrosion processes, often leading to the breakdown of the passive oxide layer on the metal surface, which can significantly accelerate corrosion. The Warburg element in this scenario reflects the diffusion of hydrated chloride ions (Cl⁻) and metal cations (Fe²⁺) through the electrolyte. The hydrated chloride species can penetrate the carbon fibre coating if it has any defects or porosity, leading to localized corrosion such as pitting. In this case, the Warburg impedance reflects the slow diffusion of these ions through the coating and the electrolyte, where mass transport becomes rate-limiting at lower frequencies. The rate-determining step would be the diffusion of chloride ions to the steel surface, especially if the coating is porous. If the carbon fibre coating is intact and effective, it may slow down the diffusion of chloride ions, thereby reducing the rate of corrosion.
The explanations added to pages 17-18 in the revised manuscript.
- The chemical composition of the solution used as a model corrosion environment is not specified. What dissolved species (component) did it contain (NaCl?) and at what concentration? Was dissolved oxygen removed, or was the solution under natural aeration conditions (the presence of dissolved oxygen affects the corrosion mechanism)?
We have added to the experimental part 2.2.5 the aggressive medium that has been employed for this study. The solution used in our experiments is 3.5 wt. % NaCl, a commonly used concentration to simulate seawater conditions, is particularly relevant for studying the corrosion behavior of materials like carbon steel in chloride-rich environments. At this concentration, the primary dissolved species are sodium (Na⁺) and chloride (Cl⁻) ions, with chloride ions playing a critical role in the corrosion process by breaking down any protective oxide layer on the steel surface, promoting localized corrosion such as pitting. The solution was kept under natural aeration conditions regarding the dissolved oxygen, meaning that oxygen was present in the solution throughout the experiment. Since we are not evaluating the oxide layer performance of the metal, which would require controlled de-aerated conditions, the use of natural aeration conditions in this case is appropriate. Our focus is more on the general performance of the coating, and we are interested in understanding how it performs in real-world conditions (where oxygen is typically present). Therefore, natural aeration conditions would likely be more representative. Coatings on steel are commonly exposed to oxygen in service environments (e.g., atmospheric exposure, seawater, etc.), making natural aeration conditions more realistic for understanding long-term performance.
- The results of this study should be compared with prototypes and analogs described in the literature.
Thank you for your comment. We acknowledge that much of the existing literature primarily investigates carbon fibre composites combined with epoxy resins or reinforced with nickel or zinc-rich coatings. These systems are widely studied for their corrosion protection properties, as epoxy and metal-reinforced coatings offer well-established benefits in terms of durability and protection. However, our study introduces a significantly different approach, which is not previously reported. Instead of using epoxy or metal-rich coatings, we have focused on Carbon Fibre-Reinforced Thermoplastic Polymer (CFRTP) composites to create multi-layered coatings. CFRTP composites are known for their unique properties, including high strength, lightweight, and processability, which distinguish them from conventional coatings.
Given these fundamental differences in both material composition and the concept of the coating system, direct comparisons with the studies on epoxy-based or metal-reinforced carbon fibre composites may not yield meaningful insights. The performance of CFRTP composites in corrosion protection is likely influenced by different mechanisms, and as such, the results of our study offer a novel perspective that cannot be directly compared to those based on other coating types.

Round 2
Reviewer 2 Report (New Reviewer)
Comments and Suggestions for Authors
The author has addressed most of my concerns and the writing has also been improved. However, there are still some issues that need further modification.
Major problems:
1) How can the author ensure the consistency and uniformity of the thickness of the steel samples after polishing with an angle grinder? (Figure 3 and Figure 4). The impact of such consistency on the bonding results cannot be ignored due to the very small thickness of 1mm. If the author cannot ensure, the subsequent strength test results will be meaningless.
Other problems:
1) The first half of the abstract still needs to be concise.
2) There are many punctuation errors, such as the use of semicolons.
3) Page4 line83: Please avoid the following mass-citation style "……coating developed in this work[4, 15-21]" , since listing those articles in an unspecific manner does not add anything specific to the article.
4) Figure3 and 4: Subheadings of subgraphs should be added. The order of SetI and SetII in Figure3 is reversed.
3) Figure14: Subheadings of subgraphs are still reversed.
4) Figure13 and 15: Suggest modifying them into bar charts with error bars.
5) Conclusions: It is suggested that the author revise the conclusion into a structural one, namely:
1) XXXX.
2) XXXX.
3) XXXX.
…………
Author Response
Reviewer #2
Comments and Suggestions for Authors
The author has addressed most of my concerns and the writing has also been improved. However, there are still some issues that need further modification.
Major problems:
- How can the author ensure the consistency and uniformity of the thickness of the steel samples after polishing with an angle grinder? (Figure 3 and Figure 4). The impact of such consistency on the bonding results cannot be ignored due to the very small thickness of 1mm. If the author cannot ensure, the subsequent strength test results will be meaningless.
We agreed with the reviewer that the consistency/uniformity of the steel-surface roughness is crucial to insure consistence test repeats. In this work, we’ve used the available tool (angle grinder) to apply some roughness on the steel-surface. However, using this method isn’t the ideal, but we did measure the surface uniformity and it is found that the level of roughness fluctuated up to 30 μm compared to the entire steel thickness 1 mm (1000 μm) i.e. the error percentage 3 % which is not bad in terms of consistency as illustrated in the following Figure.

Taking this point into consideration, we are already in the process to get rid of the hand angle grinder and looking for an alternative automated and green technology such as laser and/or plasma.
Other problems:
- The first half of the abstract still needs to be concise.
Based on the reviewer’s suggestion, the top half of the abstract has been concise (page 2, Lines 20-23), as follows: “Steel pipeline systems carry about three-quarters of the world's oil & gas. Such pipelines need to be coated to prevent corrosion and erosion. An alternate to the current epoxy-based coating, multi-layered composite coating is developed in this research. The composite coatings were made from carbon fibre-reinforced thermoplastic polymer (CFRTP) material.”
- There are many punctuation errors, such as the use of semicolons.
The authors have reviewed and corrected the punctuation typos in the text.
- Page4 line83: Please avoid the following mass-citation style "……coating developed in this work [4, 15-21]”, since listing those articles in an unspecific manner does not add anything specific to the article.
Actually, one of the previous reviewers has asked the authors to conduct a specific comparison between the polymer matrices e.g. PUR and the PPS used in the developed coating (CFRTP). That’s why we’ve provided such comparison in the manuscript (Table 1) including six references. However, based on the reviewer’s comment, we’ve done a double check for relevant references, and we found that all of them are very important to support the comparison. But we’ve excluded reference no. 4 from the citations to avoid mass-citation style.
- Figure3 and 4: Subheadings of subgraphs should be added. The order of SetI and SetII in Figure3 is reversed.
Thanks for the comment, and sorry for the mistake. The order of SetI and SetII in Figure 3 is now correct in the revised version, and the subheadings of subgraphs have been added to Figures 3 and 4 (pages 8 and 9). A new sentence has been added to lines 169-171, “Figure 4 shows a selection of metal-composite bonded coupons from the three samples UD, Biaxial and Off-axis, including images for the steel-composite interface for the Biaxial sample.”
- Figure14: Subheadings of subgraphs are still reversed.
Sorry again for the mistake. The subheadings of subgraphs are now corrected (page 22). A new sentence has been added to lines 407-408, “Figure 14 demonstrates the tested coupons of a) UD, b) Biaxial, and c) Off-axis samples.”
- Figure13 and 15: Suggest modifying them into bar charts with error bars.
Done, thanks for the suggestion.
- Conclusions: It is suggested that the author revise the conclusion into a structural one, namely:
1) XXXX.
2) XXXX.
3) XXXX.
Done, thanks for the suggestion.

Reviewer 3 Report (New Reviewer)
Comments and Suggestions for Authors
The authors have conducted a thorough and comprehensive revision of the manuscript in accordance with the provided suggestions and comments. They have given detailed explanations addressing my questions from the previous review and made appropriate corrections. I believe that the manuscript, in its current form, can be recommended for publication.
Author Response
Reviewer #1
The authors have conducted a thorough and comprehensive revision of the manuscript in accordance with the provided suggestions and comments. They have given detailed explanations addressing my questions from the previous review and made appropriate corrections. I believe that the manuscript, in its current form, can be recommended for publication.
Thank you very much for recommending the paper for publication in Polymers, highly appreciated.

Round 3
Reviewer 2 Report (New Reviewer)
Comments and Suggestions for Authors
Thank you very much for revising the manuscript adequately. The quality of the revised version has notably improved. Therefore, the article can be recommended for acceptance in the current version.
This manuscript is a resubmission of an earlier submission. The following is a list of the peer review reports and author responses from that submission.
Round 1
Reviewer 1 Report
Comments and Suggestions for Authors
The submitted paper discusses about carbon fibre-reinforced thermoplastic composite coatings for steel pipelines. The paper effectively describes the development of multi-layered coatings using CFRTP materials for protecting pipelines in the oil and gas industry. The paper has been fairly organized and can be published in Polymers. However, there are some comments that should be considered by authors before the publication of the paper.
1) The paper has not been prepared based on the policy of MDPI journals. This should be considered in the revised version of the manuscript.
2) I am not sure about the innovation of the paper. It should be explained in the introduction section.
3) In the abstract section, the authors should explain how the bonding process between the multi-layered composite coating and the carbon steel substrate influences the overall performance of the coating. It should be briefly explained.
4) In section 2.2.55, corrosion test, the authors should explain how the choice of reference and counter electrodes in the corrosion resistance evaluation influences the accuracy of the EIS measurements?
5) In pages 10 and 11, the authors should elaborate, in UD specimen, how the presence of voids and porosity affects the mechanical performance of the composite laminate?
6) In page 13, What factors could explain the need for different electrical equivalent circuits to model the EIS data for the UD coating compared to the Biaxial and Off-axis coatings?
7) In page 15, it should be elaborated how the symmetric architecture of carbon fibers in the Biaxial coating contributes to its superior corrosion resistance compared to the Off-axis and UD coatings?
8) In page 16, it should be explained, according to Fig. 12, why the Off-axis samples exhibited different failure modes.
9) In page 18, why is the adhesive agent significantly elongated in the case of the UD sample compared to the biaxial and off-axis samples?
10) In page 20, it should be explained why the UD samples exhibited the highest flexural and lap-shear strengths compared to the Biaxial and Off-axis ones.
11) The English of paper is good.
Reviewer 2 Report
Comments and Suggestions for Authors
Dear Authors! I have reviewed your manuscript "Carbon Fibre-Reinforced Thermoplastic Composite Coatings for Steel Pipelines" and I ask you to address few comments.
Introduction:
1. Lines 42-45. The words "faster and lower-cost" are to be explained giving a counterpart for comparison. Below, in lines 63-65 you list a number of existing coating systems. All of them utilize existing production infrastructure - while Carbon Fibre-Reinforced Thermoplastic coatings require a high pressure to applied INSIDE a tibe to the INNER surface of a pipe. Are you sure this technological issue can be easily answered?
2. Please, address few pharases to the issues if protection for weld seams or connection of pipes. Pipelines are to assembled from a separate tubes in the very end - how CFRT coatings are supposed to deal with connection zones?
Methods
1. Please, provide data/reference for UK producer of AS4C/PPS (TC1100) tape.
2. Ezz Steel - please, provide city and country.
3. Line 113 - looks like a part of phrase is missed.
4. Line 128 - what do you mean writing "mechanical method"? Using a grinding machine (angle hand grinder&)?
5. From Figure S7 one can conclude that CFRT coating was always in compression during flexural strength testing. Please, say it openly in the text.
Results and discussion.
You discuss F1 and F2 failure modes in lines 310-320, but these modes may appear at bottom layers where there was no composite at all - only steel. For me F3 and F6 are only modes detected. Please, answer this comment.
Round 2
Reviewer 1 Report
Comments and Suggestions for Authors
The revisions are acceptable. The paper is suggested for publication in the current format.
Author Response
Thank you very much for recommending the paper for publication in Polymers, highly appreciated.

Reviewer 2 Report
Comments and Suggestions for Authors
Dear Authors! I am partially satisfied by your answers - the main problem that for the protection of outer surface of a tube the method you study has to be compared with a variety of cheaper and faster methods - e.g. PUR foams. You did it quite superfacially - my comment on weld seams was actually ignored.
Author Response
We are sorry to know that you are partially satisfied, however, we are doing our best to make you fully satisfied.
In fact, we didn’t ignore your comments regarding:
- The comparison between the conventional coating types used for protecting steel-pipe and the proposed CFRTP coating used in this work.
- Using CFRTP composite for coating weld-seams and connectors of pipes.
We actually addressed these two points in the previous response (round 1, Comments 1 and 2).
- Regarding point I, previously we’ve provided a generic comparison between the thermoset (TS) and the thermoplastics (TP) coatings in the revised version (round 1). But it seems that a specific comparison between the polymer matrices e.g. PUR and the PPS used in the developed coating (CFRTP) is needed. Therefore, we’ve provided such comparison (including new six citations) in the revised version (round 2, lines 81-85 followed by a new table labeled as Table 1).
- Regarding point II, we’ve also answered this comment and appreciated it as it is a good idea that will be considered in the following phase of this project (prototype manufacturing) which is now in progress. However, taking this point (II) into consideration, a new paragraph including three citations has been added to the introduction section of the revised manuscript (round 2, lines 92-100).
